# Exploring the Potential of Immersive Virtual Reality in the Treatment of Unilateral Spatial Neglect Due to Stroke: A Comprehensive Systematic Review

**DOI:** 10.3390/brainsci12111589

**Published:** 2022-11-20

**Authors:** Alex Martino Cinnera, Alessio Bisirri, Ilaria Chioccia, Enza Leone, Irene Ciancarelli, Marco Iosa, Giovanni Morone, Valeria Verna

**Affiliations:** 1Scientific Institute for Research, Hospitalization and Health Care IRCCS Santa Lucia Foundation, 00179 Rome, Italy; 2Villa Sandra Institute, 00148 Rome, Italy; 3Faculty of “Medicine and Surgery”, Degree Course in Speech Therapy, University of Rome “Tor Vergata”, 00133 Rome, Italy; 4School of Allied Health Professions, Keele University, Staffordshire ST5 5BG, UK; 5Centre for Biomechanics and Rehabilitation Technologies, Staffordshire University, Stoke on Trent ST4 2DF, UK; 6Department of Life, Health and Environmental Sciences, University of L’Aquila, 67100 L’Aquila, Italy; 7Department of Psychology, Sapienza University of Rome, 00185 Rome, Italy; 8San Raffaele Institute of Sulmona, 67039 Sulmona, Italy

**Keywords:** stroke, unilateral spatial neglect, immersive virtual reality, systematic review, rehabilitation

## Abstract

The present review aims to explore the use of Immersive Virtual Reality (IVR) in the treatment of visual perception in Unilateral Spatial Neglect (USN) after a stroke. PubMed, Scopus, Embase and Pedro databases were searched, from inception to 1 February 2022. All studies that investigated the effect of IVR on USN, such as outcome in the stroke population, have been included. The current comprehensive systematic review was performed following Preferred Reporting Items for Systematic Reviews and Meta-Analyses (PRISMA) recommendations and was registered in the PROSPERO database [CRD42022311284]. Methodological quality was assessed through JBI critical appraisal tool. A total of 436 articles were identified through the database searches. A total of 10 articles, with a heterogeneous study design, which involved 77 patients with USN with low-to-moderate methodological quality, have been selected. Five out the included studies tested usability of IVR for assessed or treated visual perception deficits in USN, comparing the results with 134 healthy subjects. In the rest of studies that tested IVR such as treatment, three showed statistical positive results (*p* < 0.05) in visual perception outcome. To date, the literature has suggested the potential benefits in the use of IVR for the treatment of visual perception disorders in USN. Interestingly, IVR motivates patients during the rehabilitation process improving compliance and interest. The heterogeneity in the studies’ design and in IVR treatments indicate the need of future investigations in the consideration of potentiality and low-cost of this technology.

## 1. Introduction

Stroke is among the most common causes of disability worldwide and nowadays there are more than 80 million people who have survived a stroke [1]. After a stroke, people can experience two possible types of impairment or disability, conventionally named motor disability (including walking difficulties, problems with coordination and balance, hemiparesis, or hemiplegia) and cognitive impairments (including aphasia, memory, and visual-spatial and executive functions impairments), which are strictly intertwined each other [2]. One of the most common post-stroke cognitive impairments, which manifests in half of people who experience a stroke, is Unilateral Spatial Neglect (USN) [2,3]. USN can be defined as a deficit characterized by person’s failure to be aware of stimuli occurring on the side contralateral to the cerebral lesion, which results in the inability to report and respond to stimuli from this part of their visual field [4,5,6]. As USN is frequently associated with a lesion in the right hemisphere, people often have a left visual field deficit. This deficit is also often related to extrapersonal space [7], but it could also affect personal space [8]. Furthermore, USN is associated with poorer functional outcomes such as limited independence in daily tasks, increased risk of falls, longer hospital stays, and reduced likelihood of home discharge [9]. The patient with USN conventionally receives a pencil-paper training based on visual-scanning, reading, and copying, copying of line drawings, and verbal description of a scene [7]. With ongoing advancements in technology, new high-tech innovations, such as Virtual Reality (VR), have been introduced to stroke rehabilitation and may offer a supplementary platform for promoting physical and cognitive recovery after stroke [10]. VR is defined as a computer-based, multisensory, stimulating, real-time and interactive environment, where the individual is engaged in activities recreating real-world objects and/or events [11] This advanced technology allows people after experiencing a stroke to interact in a safe and controlled way with engaging environments having real-life features [2]. VR should be more than a simple display of digital images as a computer videogame, but it should be able to bring the observer inside a 3D Virtual Environment that could be explored and that should respond in real time to the movements of the subject in a naturalistic way [12]. Despite this, in clinical settings, often serious exergames at the basis of video-game based therapy are improperly referred to as “non-immersive” virtual reality. For the sake of clarity, and for being consistent with the clinical scientific literature, we have used this terminology in this review. In fact, in the clinical literature, VR has been classified as “non-immersive” or “immersive”, depending on the extent to which the user is isolated from the physical environment when interacting with the virtual environment [13,14]. A combination of technologies, including a head-mounted display (HMD), headphones with sound/music and noise reduction, a rumble pad, joystick, or other devices for manipulation/navigation of the virtual environment can be used to make the VR experience immersive [15]. HMD allows for the user to be fully surrounded by the virtual environment and effectively isolated from the physical reality [16]. IVR has also been demonstrated to stimulate motivation and the feeling of entertainment [17]. Therefore, VR-based cognitive rehabilitation programs have the potential to boost patients’ motivation and, as a result, reduce attrition rates [18]. IVR rehabilitation treatment may become a new option for rehabilitation after stroke [19]. Physiotherapeutic interventions based on IVR have shown positive effects in patients with USN after suffering a stroke [20]. Recent systematic reviews with meta-analysis have found VR (alone or combined with traditional treatment) to be a promising therapy for USN [2,3,21]. Although the use of VR has been investigated, studies exploring Immersive VR (IVR) have not been systematically summarized in the context of a systematic review. The aim of the current review is to summarize most common features of the IVR systems used in neurorehabilitation and their effects on reducing the visual field and attention disorders related to unilateral spatial neglect after stroke.

## 2. Materials and Methods

The current systematic review was performed following Preferred Reporting Items for Systematic Reviews and Meta-Analyses (PRISMA) recommendations [22] and was registered in the PROSPERO database (ID 311284). A literature search on several electronic databases (PubMed, Scopus, Embase, and PEDro) was conducted from inception to 1 February 2022. We combined MeshTerms and free-terms as keywords “((anterior cerebral artery stroke) OR (cerebral stroke) OR (stroke)) AND ((neglect) OR (hemisensory neglect) OR (hemispatial neglect) OR (sensory neglect) OR (perceptual disorder)) AND ((virtual reality) OR (VR) OR (exergaming) OR (immersive virtual reality)). We selected articles meeting the following inclusion criteria: (1) a hemispatial neglect population; (2) visual perception or visual attention such as primary or secondary outcome; (3) immersive virtual reality; (4) English language. Exclusion criteria were: (1); not a stroke population or a mixed sample; (2) full text not available; (3) conference paper. All results were screened simultaneously and independently by two reviewers (VV and IC). At the end of the process, in the event of no agreement, a third reviewer (AMC) was consulted. Subsequently, both reviewers independently assessed the full text of the selected articles. The following information from the studies was extracted: study authors and year of publication; description of the sample (age and sex); study design (frequency and duration of treatments); presence of a control group; IVR environment characteristics and VR tasks; study objectives; outcome measures; results and conclusions. Data of the selected studies are presented in a synoptic table (Table 1). Methodological quality of the individual studies was assessed with Joanna Briggs Institute critical appraisal tools battery (JBI) [23]. JBI is used to evaluate the trustworthiness, relevance, and results via a specific tool for each study design and is useful in the case of comprehensive reviews with heterogeneous design. Risk of Bias was assessed by two independent reviewers (IC and AB). Potential discrepancies in quality assessment were resolved through consensus or through discussion with a third reviewer (AMC). 

## 3. Results

A total of 436 articles were found. After duplicate removal (147), 289 articles were screened. After screening of titles and abstracts, 256 articles were excluded, because they did not meet the inclusion criteria. A total of 33 full-text articles were examined. As 23 studies were excluded during full-text check, 10 articles [20,24,25,26,27,28,29,30,31,32] were considered eligible for the systematic review. Flow-chart of studies screening is available in Figure 1.

### 3.1. Population

The included articles involved a total of 77 patients (51% male) with USN as a consequence of a stroke; the right hemisphere was affected in around 93% of the cases. The mean age of the stroke patients was 57.83 ± 10.74 years (all results are reported by average ± standard deviation). Five out of the ten studies selected, validated, or compared the data of IVR protocols with 134 healthy controls. The intervention groups were composed of 63 patients with USN with a mean age of 47.70 ± 9.10 years. Only one study [24] reported a control group composed of stroke patients with USN. The sample of this control group was 12, with a mean age of 61.58 ± 9.99 years. None of the included studies reported significant differences in demographic characteristics between groups (stroke or healthy). The selected studies did not report any drop-out, all individuals finished the training, and post-intervention evaluations were analyzed on the totality of the participants. Of the ten included articles, five studies [20,24,28,30,31] reported the cognitive status of the patients through the Mini Mental State Examination (MMSE). The mean MMSE score in the experimental groups was 26.33 ± 2.10, while it was 28.20 ± 1.68 in the control groups. Clear inclusion criteria were reported by all except three studies [25,28,30]; whereas the exclusion criteria were clearly defined and reported in only three studies [20,24,31]. A synoptic table with complete studies’ data is available in Table 1.

**Table 1 brainsci-12-01589-t001:** Synoptic Table of included studies.

Author, Year [Ref.] (Location)	Study Design (N Group)	Participants N, (Gender) Age ± Sd Side of Stroke (R, L)	Protocol (Frequency and Duration)	IVR System Environment	Outcome Measurements	Results (*p*.Value) ^a^
Choi et al., 2021 [24] (Republic Of Korea)	RCT (2)	IVR:12, (5 M, 7 F) 63 ± 10 (11 R, 1 L) Ctrl:12, (6 M, 6 F) 61.58 ± 10 (10 R, 2 L)	IVR: VR task with unaffected hand. (20 sessions of 1 h, for 5 days/week) Ctrl: Structured visual tracking, reading, and writing, drawing, and copying, and puzzles. (12 sessions of 30 min for 3 days/week).	HMD (Oculus Rift Development Kit 2, Facebook Inc., Menlo Park, CA) and Windows Runtime 0.8.0-β.	LBT; CBS; MBI; MVPT-V; and head tracking.	IVR showed significantly greater improvements in the LBT (*p* = 0.02) *, in the visual perceptual test (*p* < 0.02) * and in the horizontal head movement of rotation degree (*p* = 0.007) * and velocity (*p* = 0.001) *.
Yasuda et al., 2017 [20] (Japan)	Pre-post Design (1)	10, (6 M, 4 F) 45 ± 8.5 (10 R)	IVR: Far/near space training with VR visual searching/reaching tasks. (1 session of ~30 min)	HMD (Oculus Rift Development Kit 2, Oculus VR Inc., Irvine, CA, USA), a motion-tracking device (Leap Motion, Leap Motion Inc., San Francisco, CA, USA), and a PC.	BIT (Line cancelation task; Star cancelation task; Letter cancelation task; and LBT)	BIT scores obtained pre-and post-VR program revealed an improvement in far space neglect (*p* = 0.002) ** but not in near space neglect (*p* = 0.18) **. This effect for far space neglect was observed in the cancelation task (star and letters), but not in the LBT.
Castiello et al., 2004 [25] (United Kingdom)	Case Control (2)	IVR:6, (3 M, 3 F) 71.8 ± 3 (6 R) HV:6, (NR) 73 (NR)	IVR: 3 VR tasks with location, reaching and grasping activities. (3 sessions of 60, 120 and 20 trials, respectively). HV: Same protocol of the IVR. (Same duration and frequency).	A data glove (Virtual Reality; Fifth Dimension Technologies, Irvine, CA) and a PC.	Motor task; and sensory task.	An increase in the % of correct responses for the left trials was observed between the 3rd session with respect to the 1st session (*p* < 0.001) **, and in the % of correct left responses after having experienced the left-incongruous trials (*p* < 0.001) *^b^.
Heyse et al., 2022 [26] (Belgium)	Validation Paper (2)	IVR:4, (NR) NR (3R, 1L) HV:4, (NR) NR NA	IVR: 4 VR tasks: (1) “Assessment”, (2) “Scales”, (3) “Memory” and (4) “Free-to-Play”. (6 sessions of 30 min, 3 days/week). HV: same IVR protocol. (1 session of 20 min)	HMD and gloves or controllers.	CBS; TAP. IVR tasks evaluated were: “Assessment”; “Scales”; “Memory”; “Free-to-Play”.	Patients increasingly corrected their head direction towards their neglected side. Patients responded to triggers and performance results could be clearly differentiated between clinical and non-clinical users.
Baheux et al., 2007 [27] (Japan)	Validation Paper (3)	IVR:2, (1 M, 1 F) 72 ± 1 (2R) HV1:22, (13 M, 9 F) senior: 73.3 ± 4.6 young: 25.3 ± 3.6 (NA) SP2:22, (19 M, 3 F) senior: 70.5 ± 9.2 young: 23.2 ± 2 (NA)	Virtual line bisection test with virtual paper and pencil tests (NR)	Eye-tracking device, a haptic device and a Sharp Mebius PC-RD1-3D notebook. This notebook has a stereoscopic display that does not require the wearing of stereo glasses. A Phantom Omni made by Sensable, was used to interact with the virtual world.	Eye-gaze patterns and performance.	Patients and healthy simulated patients had similar eye-gaze patterns. However, while the reduced visual field condition had no effect on the healthy simulated patients, it had a negative impact on the patients.
Kim et al., 2007 [28] (Republic Of Korea)	Validation Paper (3)	IVR:10, (5 M, 5 F) 51.4 ±16.3 (10 R) HV1:20, (2 M,18 F) 59.8 ± 5.0 (NA) HV2:20, (17 M, 3 F) 29.7 ± 2.3 (NA)	Virtual LBT; virtual cancellation test, and “traffic light” game (search the vehicle by rotating the head). (NR)	An HMD with a 3 DOF’s head tracking was used to measure subject head movement in virtual environment. A PC.	Deviation angle; reaction time; right reaction time; left reaction time; visual cue; auditory cue; failure rate of mission.	Has been found that it is possible to reduce the asymmetry between left and right side by training patients to compensate for contralateral visual sites.
Hagiwara et al., 2018 [29] (Japan)	Case Series	4, (NR) 64.0 ± 11.2 (NR)	Search and read a number with a series of 4 command: (1) display the clue stimulation, (2) blackout the surrounding environment, (3) move the clue stimulation, and (4) remove the blackout. (1 session of 10 repetitions).	HMD (Oculus Rift CV1, Oculus VR., Inc.) and a PC. A tracking sensor provided to acquire information on the position and rotation of the patient’s head.	Apple Test; LBT.	Patients showed reduced error ratio on the Apples Test. The percent deviation in the LBT for all of the patients tended to be reduced (Descriptive analysis).
Kim et al., 2004 [30] (Republic Of Korea)	Validation Paper (3)	IVR: 12 (8 M, 4 F) 54.9 ±17.4 (NR) HV1: 20 (15 M, 5 F) 29.5 ± 2.5 (NA) HV2: 20 (15 M, 5 F) 59.9 ± 6.1 (NA)	“Track a ball” (1 session).	The VR System consisted of a Pentium IV PC, DirectX 3D Accelerator VGA Card, Head Mount Display (HMD, Eye-trek FMD-250W) and a 3 Degrees Of Freedom Position Sensor (Intertrax2).	The deviation angle; the no attention time; the scanning time; the number of cues; the failure rate of mission; MVPT; CPM; and WMS.	The six outcome parameters showed a significant difference between patient group and normal group when using this program as an assessment tool.
Yasuda et al., 2008 [31] (Japan)	Case Report	1 M 76 (1 R)	Far and near space training. (30 sessions of 30 min for 5 days/week).	HMD (Oculus Rift Development Kit 2, Oculus VR, Irvine, California, USA), a motion-tracking device (Leap Motion, Leap Motion, San Francisco, California, USA) and a PC.	Line cancellation test; LBT; and CBS.	Positive effects of the IVR program for far space neglect are suggested (descriptive analysis).
Smith et al., 2007 [32] (Canada)	Case Series	4, (4 F) 49.3 ± 5.8 (2 R)	Ten trials of VR games (“Birds and Balls”, “Soccer”). (6 sessions of 1 h for 1 day/week).	The Mandala Gesture Xtreme VR system and Interactive Rehabilitation Exercise software.	BIT; and Bells Test.	A positive effect has been observed in all patients, seen differently in the Bells test and in the BIT (descriptive analysis).

Abbreviations: BIT = Behavioral Inattention Test; CBS = Catherine Bergego Scale; RCT = Randomized Controlled Trial; Ctrl = Control Group; CPM = Raven’s Colored Progressive Matrices; DOF = Degrees of Freedom; HMD = Head Mounted Display; HV = Healthy Volunteers; IVR = Immersive Virtual Reality; L = Left; LBT = Line Bisection Test; MBI = Modified Barthel Index; MVPT = Motor-Free Visual Perception Test; MVPT-V = MVPT-Vertical Version; NA = Not Applicable; NR = Not Reported; PC = Personal Computer; R = Right; RCT = Randomized Controlled Trial; s = seconds; sd = standard deviation; SP = Simulated Patients; USN = Unilateral Spatial Neglect; VR = Virtual Reality; WMS = Wechsler Memory Scale; ^a^
*p*.value not reported for the validation studies; * between-group analysis; ** whiting group analysis; *^b^ between-group analysis compared with healthy subjects.

### 3.2. Intervention Characteristics

In the selected studies, the IVR training was characterized by a great variety of tasks (a complete description of each IVR task is reported in Table 2). Two studies used a virtual version of the line bisection test as intervention protocol [27,28]. In six studies, the intervention protocol was composed of more than one task/action [25,26,28,29,31,32]. Three studies made use of an IVR program [20,24,30]. The IVR tasks used in the intervention protocols have the role of training near and far space. Furthermore, the patient’s movements are reflected in the IVR space with the appropriate sensors. These exercises aim to guide the patient’s attention to the neglected side of the target object. The different systems turn off the surrounding stimuli (i.e., other stimuli around the target) through the blackout in the VR environment, as shown in the study by Hagiwara et al. [29]. Session frequency and duration were different in almost all of the studies; only two studies [24,32] had the same duration of the session (one hour), but their protocols were different. Four studies [27,28,29,30] did not report the frequency and the duration of the sessions. The duration of the sessions ranged from 5 [20] to 30 min [24,31]. All intervention groups performed an IVR protocol, although the included studies used different immersive systems. Most of the studies [20,24,26,28,29,30,31] used an HDM; one study [25] used a glove, another one [27] used a haptic device and another one [32] used the Mandala Gesture Xtreme VR system and the Interactive Rehabilitation Exercise (IREX) software. 

**Table 2 brainsci-12-01589-t002:** IVR tasks description.

Author, Year	IVR Task	Task Description
Choi et al., 2021 [24]	1. 10 IVR Applications	Participants wear Oculus Rift DK2 and Leap Motion and are seated in a chair to perform 10 different applications (e.g., “Blocks”, “Element L”, “Warlock”, “Laser”, “Pinch Draw”, “RPS island”, “VR table tennis”) from Oculus share and Leap Motion developers. A visual searching task in the VR space for far space training is used. A virtual screen was located at 15-m distance and seven visual stimuli were placed on the screen. Visual stimuli flashed consecutively for 6 s each, from the right to the left of the screen. The task requires the patient to extend their hand in VR to touch each object (object turns red) in order from right to left. A visual searching task in the VR space for near space training is used. The task requires the patient to orally identify each flashing object. Three objects are placed on the table in the VR space. The task requires the patient to extend their hand in VR to touch each object (object it turns red) in order from right to left. Two types of tasks, “sensory” and “motor,” are performed within the real or virtual environment. In the sensory task, subjects are required to report the location in which the object appears, whereas for the motor task, the subjects are required to reach and grasp the object. For all tasks, the order of stimulus presentation is counterbalanced across participants. Subjects are instructed to reach for the real object located at one of the three predefined locations within the real environment while simultaneously being able to view only a real-time representation of the virtual hand. Subjects are instructed to perform the “sensory” task as in point (5). This measures the effect of the manipulation on performance of the sensory task. The instrument keys are set up symmetrically and each one is supported by a small ball that starts up in the air and slowly falls towards the keys. Patients are instructed to avoid letting the balls touch the keys. When a ball is close to the surface of a key, it turns green, indicating that the patient can then hit the corresponding key to send the ball back up into the air. When the ball touches the key, it turns red. In this task, the patient has to play scales on the xylophone, i.e., the patient has to sequentially hit each key in order, starting from their non-neglected side towards their neglected side. The therapist tells the patient to memorize a sequence of notes that is shown to them and then repeat this sequence. This task starts from the lowest difficulty level, showing a sequence of one note, each time increasing the difficulty. When the patient plays an incorrect key, the sequence is shown again; when they play the correct key, positive feedback is given. In this task, the objective is to provide the patient with some ‘cognitive downtime’ by letting them play freely. The game mechanics are the same as points (7), (8) and (9). The use of small balls falling to the surface of the keys again triggers the active exploration of the patient, urging them to explore their entire environment to avoid any balls hitting the keys. The virtual LBT consists of marking the midpoint on nine lines presented one at a time. The lines can have three different lengths (50 mm, 100 mm and 150 mm) and three different positions (left side, centered and right side). The trials are randomized, and the origin of the haptic device was shifted by 25 mm to the right to avoid judgments based on the body midline. The virtual LBT is performed in the normal condition and with a visual field reduced to a round area (in order to decrease the effect of the USN). This round area is constantly moving back and forth along the line. Patients performed an IVR game in which a traffic light changed red into green. Patients had to search the vehicle rotating their head and had to push the mouse button to close the mission. The therapist operates the system with a series of four actions, which are as follows: 1. display the clue stimulation, 2. blackout the surrounding environment, 3. move the clue stimulation, 4. remove the blackout. After these actions, the therapist asks the patient to read the four-digit number on the green panel. In the main task, the subject must detect the ball using their gaze (moving a small cross according to the subject’s head motion). The subject must maintain their gaze on the ball during the ball’s movement time. The patient performs a visual search task in the VR space. The patient performs a reaching task in the VR space. The balls appear from various directions on the screen and the patient bursts the balls with his/her hands to change them into birds. A soccer ball comes up on the screen and the patient stops the ball from going in the goal with his/her hands by playing the role of a goalkeeper.
Yasuda et al., 2017 [20]	2. “Visual searching”; 3. “Virtual object for reaching tasks”.	
Castiello et al., 2004 [25]	4. “Baseline task”; 5. “Real/virtual task”; 6. “Sensory task”.	
Heyse et al., 2022 [26]	7. “Assesment”; 8. “Scales”; 9. “Memory”; 10. “Free-to-Play”.	
Baheux et al., 2007 [27]	11. “Virtual Line Bisection Test”.	
Kim et al., 2007 [28]	12. “Traffic Light”	
Hagiwara et al., 2018 [29]	13. A four actions task	
Kim et al., 2004 [30]	14. A task with the gaze on a ball	
Yasuda et al., 2008 [31]	15. “Far space training”; 16. “Near space training”.	
Smith et al., 2007 [32]	17. “Birds and ball”; 18. “Soccer”	

Abbreviations: VR = Virtual Reality; IVR = Immersive VR; LBT = Line Bisection Test; USN = Unilateral Spatial Neglect.

### 3.3. Comparison

Of the included studies, six [24,25,26,27,28,30] used a comparison protocol to verify the effects of the IVR training. Five studies [25,26,27,28,30] used the same protocol of the intervention group. Only one study [24] used a different comparison protocol than the intervention one in which various tasks were performed, such as structured visual tracking, reading, and writing, drawing and copying, and puzzles. Four studies [20,29,31,32] did not include a control group in their study design. The studies that used a control group could be considered adequate as a comparison with the condition being tested as they used specific neglect exercises. The duration and frequency of the training sessions were mainly heterogeneous. Only one study [25] used a control training with the same duration and frequency of the intervention group training. Only one study [26] used a different duration and frequency of the control group training compared to that of intervention, with a single session of around 20 min. Three studies [27,28,30] did not report the frequency and the duration of the sessions.

### 3.4. Outcome

The selected studies used different measures for the primary outcome to evaluate the effects of IVR training. Two studies [24,26] used the Catherine Bergego Scale (CBS), a functional scale that allows for the detection of the presence and the degree of abandonment during the observation of everyday life situations. Two studies [20,32] used the Behavioral Inattention Test (BIT), which is composed of conventional sub-tests and behavioral sub-tests for the assessment of neglect. Two studies [29,31] used the line bisection test, a conventional sub-test of the BIT, that evaluates peri-personal neglect. In the remaining studies, different measures were used. Two studies [28,30] investigated seven parameters (deviation angle; reaction time; right reaction time; left reaction time; visual cue; auditory cue; failure rate of mission). One of the two studies [30] also calculated the ratio of the right to left scanning time. One study [25] used “Baseline task”, “Real/virtual task”, “Sensory task”. Virtual objects, subject’s interactions, eye-gaze and the positions of the marked mid-points were recorded in one study [27]. None of the included studies used secondary outcome measures. The studies included used different scales or tests to assess the USN. The most used were the BIT, used in four studies [20,29,31,32], the CBS used in three studies [24,26,31] and the Motor-Visual Perception Test (MVPT) used in two studies [24,30]. Other scales/tests used were the Raven’s Colored Progressive Matrices (CPM), the Wechsler Memory Scale (WMS), and the Line Bisection Test (LBT) which is extracted from the BIT, and the Bell’s test. Only three studies [25,27,28] did not evaluate the USN condition with a scale or a specific test. Follow-up assessments were performed only in two studies [30,31]. In the first study [30] although the degree of neglect increased slightly at the 3-month follow-up, rather than in the last training, and the authors considered this system to be effective in USN. The second study [31] reported neither duration nor results of the declared follow-up.

### 3.5. Risk of Bias

We used the JBI checklists to assess the risk of bias of the studies included in the review. For only one RCT study [24] was the “Checklist for the Randomized Controlled Trial” tool used [33]; for only one case report [31] was the “Checklist for case reports” tool used [34]; for two case-series [29,32], the “Checklist for case series” tool was used [35]; for only one case-control [25] was the “Checklist for Case Control Studies” used [36]; finally, for the other studies [20,26,27,28,30], “Checklist for qualitative research” was used [37]. Overall risk of bias in the RCT [24] was low. The two negative items concerned the follow-up and the blind of those delivering treatment. In this type of study, the therapist must necessarily know of the IVR treatment assigned to the patient groups. All of the methodological studies [20,26,27,28,30] have a low risk of bias. The high overall risk of bias in one [32] of the two case-series was due to the lack of a complete or consecutive inclusion of the participants and to the statistical analysis, which was not appropriate for the type of data collected. The moderate risk of bias in the case-control study [25] was attributable to the lack of comparability between groups and to the absence of the same criteria used for the identification of cases and controls. Finally, the case report [31] had a low risk of bias. Overall, the risk of bias of the studies evaluated was low/moderate, although the general methodological quality was unsatisfactory. A summary of the risk of bias assessment is presented in Figure 2.

## 4. Discussion

This systematic comprehensive review aimed to explore the effect of immersive virtual reality in the treatment of visual perception deficits due to unilateral spatial neglect in stroke patients. The screening of the literature provided 10 studies involving 77 subjects with USN and 134 healthy subjects. We acknowledge that IVR is a topic of interest not only for rehabilitation, but also in the assessment of USN. VR is an emerging technology and its related devices have recently been developed. IVR is considered as a safe and inexpensive option for treatment [38], and it stimulates patients’ motivation by adding gaming factors in a safe virtual environment [29]. 

First of all, our review highlighted, as in the clinical studies about the use of IVR for treating USN, small samples of patients which had been enrolled to obtain solid conclusions. In 10 studies, 211 subjects were enrolled, and about two-thirds of them were healthy subjects included for testing the system and/or providing physiological baseline for the data. Despite the risk of bias being quite low, we should note that a few studies reported a control group performing conventional therapy. This could influence the interpretation of results, especially because USN is a deficit that partially improves also without therapy [7]. Furthermore, there was a wide variety of protocols and assessed outcomes.

In the analyzed studies, no relevant severe adverse events were reported. However, in other IVR experiences, the literature reported slight symptoms such as dizziness, nausea, sore eyes, and disorientation [38]. In their review, Tsirlin et al. [39] highlighted some characteristics of VR technologies that should be considered for future VR applications in this field. The most important is the ergonomic aspect of VR tools, as people post-stroke have specific needs that need to be considered, such as limited mobility [39]. Five of the studies included in this review [20,24,25,31,32] specified that the training was performed seated in a wheelchair or in a chair. Symptomatology, limitation to maintaining the upright position, and limitation in mobility suggests the use of IVR in a supervised environment such as a clinical setting. Despite the potential in-home implementation, current evidence reports use in a controlled setting, and safety assessments are needed before tested IVR technology in a domestic environment.

A second important challenge that may limit VR implementation in the assessment and rehabilitation of USN is the high costs associated with designing and testing a technological system, in front of the reduced costs of the hardware [39]. However, this aspect has not been taken into consideration in the studies included in this review. What emerges from our study is that, although most studies have used an HMD, there is a vast heterogeneity of the IVR systems used. To date, studies [24,26,29,32] have shown that through IVR treatment, it is possible to find significant improvements in USN deficit. Patients undergoing IVR training showed an increase in visual perception and head movements immediately after training and after three months [28]. These findings are in line with other results achieved by previous studies [40,41,42] that have investigated the effect of VR training in the same population. Six of the included studies used moving stimuli (visual, auditory or both) to guide the patient’s attention to the neglected side of the target object [29]. In line with this, the use of moving stimuli may be crucial to modulate and drive patients’ visual attention to the left side of the space [2]. Some studies used a Kinect system that can provide an avatar representation of the upper limb which allowed subjects to interact with the virtual scenarios. This coupled approach allows for the creation of task-oriented stimuli, such as reaching and grasping and objects, unifying motor, and explorative skills. Interestingly, compared to conventional rehabilitation methods, IVR motivates patients during the rehabilitation process, improving compliance and interest [24,26,30]. The IVR environment involves a high degree of presence and immersion for the user, thus producing ecological and relevant exercises. Furthermore, the possibility of creating personalized, ecological, and repetitive treatments could maximize the results. For the assessment of outcomes, the studies used a wide variety of different scales and tasks. To contrast this high variability, ladders designed solely for negligence, such as CBS and BIT, should be used. However, it was noted that only five studies used them as outcome rating scales, reporting a non-change in patients’ scores after intervention [24,31] while in one study [24], a more significant correlation with the CBS score became evident. While one study [32] using the BIT test showed a reduction in USN symptoms, in one study [29] no statistically significant differences were observed between pre- and post-test. The duration of the sessions differed in each study. In one study [28], although the duration and frequency of the sessions were not reported, the 3-month follow-up showed that the effect of the training remained. From this, it may be inferred that IVR has potentially effects that last over time, but further investigations are needed to confirm this. The selected studies used various protocols and non-standard therapies. Nevertheless, all studies demonstrated the effectiveness of IVR therapy. Despite some limitations, the use of IVR seems to be a promising and effective method for post-stroke treatment in patients with USN.

### 4.1. Limitations

There are several limitations to this study. First, the findings of this review are mainly based on studies with research designs other than RCTs, due to the paucity of RCTs available to date. Second, the difference between the studies in terms of study design, interventions, type, and techniques of IVR did not allow for a meta-analysis to be conducted. Lastly, the selected studies had a very limited sample size, so future studies with larger sample sizes are needed. 

### 4.2. Future Perspectives

To confirm our current observations, other well-designed studies are needed. Future RCTs design studies with representative populations can clarify the effect of IVR in USN treatment. Particular attention should be paid to the duration and frequency of the treatment, in both experimental and comparison group, and to adequate follow-up evaluation choices. 

## 5. Conclusions

The present comprehensive review showed the potential benefit of the use of IVR for the treatment of USN through audio-visual dynamic stimuli (from the right to the left side). An improvement in the visual perception and head movement, with good compliance, was frequently reported. Nevertheless, high protocols’ heterogeneity, unsatisfactory methodological quality, and limited sample size were observed, necessitating further investigation to confirm the potential benefit of IVR in treatment of visual perception and attention disorders following a stroke. The wide potentiality given by the IVR seemed to bring to a large variability of protocols, with some outcomes strictly intertwined with the VR protocol. There is the need to define the neuroscientific criteria behind the development of VR environments and tasks, and to at least have a common approach within these criteria. At the same time, the assessments should be based on clinical scales independent by the adopted IVR, even if the analysis of kinematic data that can be measured by IVR systems could be helpful for monitoring the ongoing improvements of the patients. 

## Figures and Tables

**Figure 1 brainsci-12-01589-f001:**
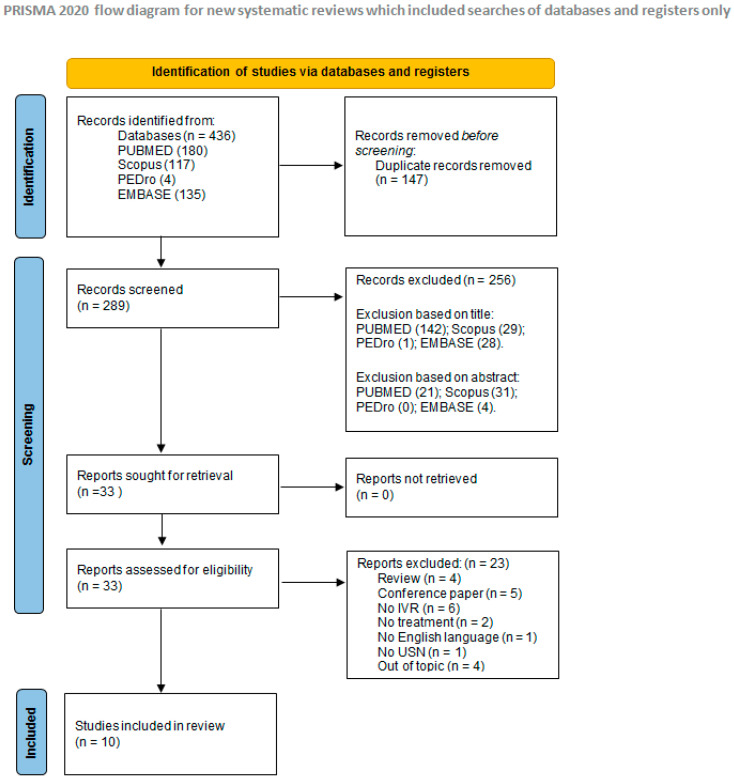
PRISMA flow Diagram of studies selection process [22].

**Figure 2 brainsci-12-01589-f002:**
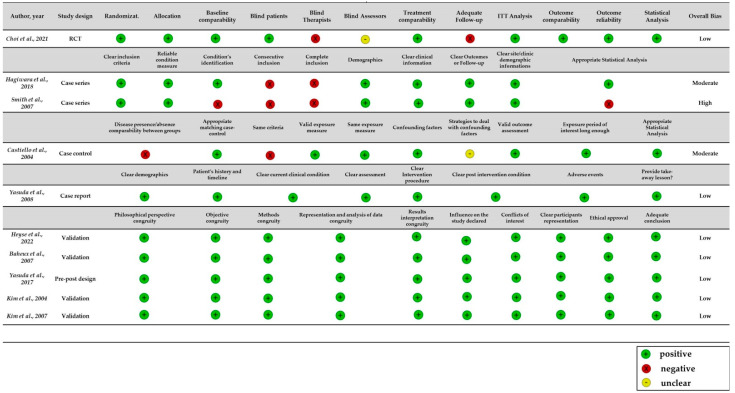
Methodological quality evaluated with the Joanna Briggs Institute Critical Appraisal tools.

## Data Availability

The datasets used during the current study are available from the corresponding author on reasonable request.

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
