# Peer review of "Exploring the Potential of Immersive Virtual Reality in the Treatment of Unilateral Spatial Neglect Due to Stroke: A Comprehensive Systematic Review"

_brainsci, 2022, doi:10.3390/brainsci12111589_

Round 1

Reviewer 1 Report

Title/Abstract: 

The title could suggest that the review is exploring the efficacy of IVR as later suggested in line 73. The abstract could include some statistical findings such as P values ranged from P=.87 to P ≤.001. It is also unclear what outcomes were measured other than ‘the treatment of unilateral spatial neglect.

Intro:

The aim of the systematic review is vague and unclear. One would normally expect a systematic review to explore the effectiveness of an intervention and stipulate the outcome i.e. function. This does not clearly do this.

Materials and Methods: 

I’m unsure why the authors have only used “(anterior cerebral artery stroke) OR (cerebral stroke) OR (stroke)), I would have expected additional text terms such as Cerebrovascular accident, CVA, hemorrhagic stroke.

The authors have not stated what outcome measures were included or excluded. I would have liked some justification for the risk of bias tool used as the JBL is not considered gold standard.

Results:

The summary table does not contain key information such as the numbers recruited (N = , n=), the results and the outcome measures used. The paper does not adequately report the risk of bias. I would expect a systematic review to fully critique the studies and the domains of the risk of bais. 

Discussion / Conclusion:

A limited discussion and conclusion are offered. The authors could have written a pragmatic description of the practicalities of using VR such as where and how users could access the systems and the motor learning principles underpinning the intervention. There is little reflection of the outcomes of the studies and in which domain i.e. function the outcomes benefit.

Author Response

All authors thank the reviewers and editors for taking your time to improve the manuscript. Below are the authors 'responses to reviewers' comments (blue text color). We hope to have solved all concerns in the current version of the manuscript.

REVIEWER 1

Title/Abstract:

The title could suggest that the review is exploring the efficacy of IVR as later suggested in line 73. The abstract could include some statistical findings such as P values ranged from P=.87 to P ≤.001. It is also unclear what outcomes were measured other than ‘the treatment of unilateral spatial neglect.

R: Thanks for the interesting observations. We modified the title in line with your suggestion “Exploring the potential of immersive virtual reality in the treatment of unilateral spatial neglect due to stroke: a comprehensive systematic review”.

We now include in the current version of the manuscript the p.value in the between-group and within-group analysis in new table 1 and in the abstract: “In the rest of studies that tested IVR such a treatment, three out of these showed statistical positive results (p < 0.05) in visual perception outcome”.

We clarify the aim of our systematic review here “The present review aims to evaluate the use of immersive virtual reality in the treatment of visual perception in unilateral spatial neglect (USN) after a stroke.” and in the introduction section of revised manuscript ”The aim of the current review is to summarise most common features of the IVR systems used in neurorehabilitation and their effects on reducing the visual field and attention disorders related to unilateral spatial neglect after stroke.”.

Intro:

The aim of the systematic review is vague and unclear. One would normally expect a systematic review to explore the effectiveness of an intervention and stipulate the outcome i.e. function. This does not clearly do this.

R: Thanks for your comment, we change the sentence to clarify the aim of our review. We changed the past sentence “The aim of the current review is to investigate for the first time the potential of immersive virtual reality in the treatment of USN after a stroke” with “The aim of the current review is to summarise most common features of the IVR systems used in neurorehabilitation and their effects on reducing the visual field and attention disorders related to unilateral spatial neglect after stroke”.

Materials and Methods:

I’m unsure why the authors have only used “(anterior cerebral artery stroke) OR (cerebral stroke) OR (stroke)), I would have expected additional text terms such as Cerebrovascular accident, CVA, hemorrhagic stroke. The authors have not stated what outcome measures were included or excluded. I would have liked some justification for the risk of bias tool used as the JBL is not considered gold standard.

R: We are grateful for this observation. We used the MeshTerm with a large tree to includes the maximum possible number of results (i.e. “Stroke” Mesh Term [C10.228.140.300.775] that includes a large spectrum of terms), but reporting in the past main document  and in the supplementary material a short form of search strategy. In the current supplementary material we reported the complete search strategy used, adding your suggested keywords.

 “Stroke” MeshTerm include a lot of terms (i.e. Apoplexy, CVA (Cerebrovascular Accident), Cerebral Stroke, Cerebrovascular Accident, Cerebrovascular Accident, Acute, Cerebrovascular Apoplexy, Cerebrovascular Stroke, Stroke, Acute, Vascular Accident, Brain), and authors probably included stroke as keyword also if in their papers they refer to another terminology (such as CVA). Anyway, we have performed a new search with suggested keywords without difference in the results, probably for the above mentioned reason.

((("infarction, anterior cerebral artery"[MeSH Terms] OR ("stroke"[MeSH Terms] OR "stroke"[All Fields] OR ("cerebrovascular"[All Fields] AND "accident"[All Fields]) OR "cerebrovascular accident"[All Fields]) OR "stroke"[MeSH Terms] OR ("stroke"[MeSH Terms] OR "stroke"[All Fields] OR "strokes"[All Fields] OR "stroke s"[All Fields])) AND ("neglect"[All Fields] OR "neglected"[All Fields] OR "neglectful"[All Fields] OR "neglecting"[All Fields] OR "neglects"[All Fields] OR "perceptual disorders"[MeSH Terms] OR "perceptual disorders"[MeSH Terms] OR "perceptual disorders"[MeSH Terms])) OR "perceptual disorders"[MeSH Terms]) AND ("virtual reality"[MeSH Terms] OR ("proc ieee virtual real conf"[Journal] OR "vr"[All Fields]) OR ("exergamers"[All Fields] OR "exergaming"[MeSH Terms] OR "exergaming"[All Fields] OR "exergame"[All Fields] OR "exergames"[All Fields]) OR (("immerse"[All Fields] OR "immersed"[All Fields] OR "immerses"[All Fields] OR "immersing"[All Fields] OR "immersion"[MeSH Terms] OR "immersion"[All Fields] OR "immersions"[All Fields] OR "immersive"[All Fields] OR "immersiveness"[All Fields]) AND ("virtual reality"[MeSH Terms] OR ("virtual"[All Fields] AND "reality"[All Fields]) OR "virtual reality"[All Fields]))) AND 1990/01/01:2022/01/31[Date - Publication]”

We clarify in the current version the outcome of investigation in the abstract, introduction and methods sections:

“The present review aims to evaluate the use of immersive virtual reality in the treatment of visual perception in unilateral spatial neglect (USN) after a stroke.” [...] ”The aim of the current review is to summarise most common features of the IVR systems used in neurorehabilitation and their effects on reducing the visual field and attention disorders related to unilateral spatial neglect after stroke.” [...] “We selected articles that meet the following inclusion criteria: 1) hemispatial neglect population; 2) visual perception or visual attention such as primary or secondary outcome; 3) immersive virtual reality; 4) English language.”

About the risk of bias, we agree with your comment, the commonly used scales like ROB tool 2 of Cochrane or the PEDro scale are suggested in systematic review and meta-analysis that included RCTs. Moreover, for our comprehensive review that included multiple study designs, the use of JBI offers the opportunity to use a fit tool for each study design without changing in the scale. We specify it in the present version of the manuscript with following changes in the first sentence of “risk of bias” paragraph “Methodological quality of the individual studies was assessed with Joanna Briggs Institute critical appraisal tools battery (JBI). JBI is used to evaluate the trustworthiness, relevance and results via a specific tool for each study design and is useful in case of comprehensive review with heterogeneity about the design.”. However, we are willing to use a different tool, if expressly requested by the reviewers or the editor.

Results:

The summary table does not contain key information such as the numbers recruited (N = , n=), the results and the outcome measures used. The paper does not adequately report the risk of bias. I would expect a systematic review to fully critique the studies and the domains of the risk of bias.

R: Thanks for the comment, according to it, we transferred Table 1 from supplementary materials to the main document to provide the required information in the main text. Furthermore, we have revised the entire table with the suggested information. About the risk of bias, please see our response to your previous comment and Figure 2 in which the risk of bias has been reported for the analysed studies.

Discussion / Conclusion:

A limited discussion and conclusion are offered. The authors could have written a pragmatic description of the practicalities of using VR such as where and how users could access the systems and the motor learning principles underpinning the intervention. There is little reflection of the outcomes of the studies and in which domain i.e. function the outcomes benefit.

R: We have improved Discussion in many paragraphs and also introduced the following new paragraphs:

First of all, our review highlighted, as in the clinical studies about the use of VR for treating USN, small samples of patients have been enrolled to obtain solid conclusions. In 10 studies, 211 subjects have been enrolled, and about 2 thirds of them were healthy subjects included for testing the system and/or providing physiological baseline for the data. Despite the risk of bias being quite low, we should report as few studies reported a control group performing conventional therapy. It could influence the interpretation of results, especially because USN is a deficit that partially improves also without therapy [Matano et al., 2015]. Furthermore, there was a wide variety of protocols and assessed outcomes.” [...] “The wide potentiality given by the IVR seemed to bring to a large variability of protocols, with some outcomes strictly intertwined with the VR protocol. There is the need to define the neuroscientific criteria behind the development of particular VR environments and tasks, and to have at least a common approach in these criteria. At the same time, the assessments should be based on clinical scales independent by the adopted IVR, even if the analysis of kinematic data that can be measured by IVR systems could be helpful for monitoring the ongoing improvements of the patients.”

Reviewer 2 Report

Title: Immersive virtual reality for treatment of Unilateral Spatial Ne- 2 glect due to stroke: a comprehensive systematic review

The present review aims to evaluate the use of immersive virtual reality in the treatment  of unilateral spatial neglect (USN) after a stroke. I recommend to summarize Results and broad the Discussion.

Main comments

In general, the manuscript is well-written. However, Introduction and Discussion could be expanded.

1. Introduction

As a review article, it would be appropriate to incluide further information about the topic.

2. Materials and Methods

Line 29: Delete “multiple” or change “several electronic databases” instead of “multiple electronic databases”.

3.Results

Poor quality of Figure 1, please add another of better quality.

Line 121: Review the total number of subjects included in the article, it comes out 211 instead of 199.

Lines 136-138: “A synoptic table with complete studies' data is available in a comprehensive Table of supplementary materials (S2)”.

I recommend to eliminate this Table from the supplementary material section and incorporate it into the text for a better reading and understanding.

Lines 140-142: “In the selected studies, the IVR training was characterized by a great variety of tasks (complete descriptions in Table 1). A description of each IVR task was reported in Table 2”.

It does not correspond to the table incorporated in the text.

First of all, the Table has no legend and no table number. I recommend to insert a legend and the corresponding number.

Second, there is only one table. It looks like two tables have been joined, but the explanation in the text has not been corrected.

Line 149: Hagiwara and colleagues [25]

Write Hagiwara et al. [25]

Line 217: finally, for the other studies [22-24, 26]

Incorporate study 17: [17, 22-24, 26]

4.Discussion

Line 226: 77 subjects with USN and 122 healthy subjects.

Review the number of healthy subjects

Line 236: Tsirlin and colleagues in their review [35]

Substitute for Tsirlin et al. [35] in their review

Line 252: (Hagiwara et al., 2018).

Replace with [25]

5. References

Line 359: review Hagiwara et al.

As a review article, it would be appropiate to increase the number of references in certain sections (Introduction and Discussion).

Author Response

REVIEWER 2

The present review aims to evaluate the use of immersive virtual reality in the treatment  of unilateral spatial neglect (USN) after a stroke. I recommend to summarize Results and broad the Discussion.

Main comments

In general, the manuscript is well-written. However, Introduction and Discussion could be expanded.

R: Thank you for your general positive judgement about our work and for these and further qualified comments that helped us to improve our review. As reported below, we have also expanded Introduction and Discussion, in particular the former with a more detailed description of the pathology, and the latter with a deeper analysis of the findings.

Introduction (please see our response to your next comment) and Discussion have been expanded. In particular many paragraphs of Discussion has been revised the following ones added ex-novo:
First of all, our review highlighted, as in the clinical studies about the use of VR for treating USN, small samples of patients have been enrolled to obtain solid conclusions. In 10 studies, 211 subjects have been enrolled, and about 2 thirds of them were healthy subjects included for testing the system and/or providing physiological baseline for the data. Despite the risk of bias being quite low, we should report as few studies reported a control group performing conventional therapy. It could influence the interpretation of results, especially because USN is a deficit that partially improves also without therapy [Matano et al., 2015]. Furthermore, there was a wide variety of protocols and assessed outcomes.” [...] “The wide potentiality given by the IVR seemed to bring to a large variability of protocols, with some outcomes strictly intertwined with the VR protocol. There is the need to define the neuroscientific criteria behind the development of particular VR environments and tasks, and to have at least a common approach in these criteria. At the same time, the assessments should be based on clinical scales independent by the adopted IVR, even if the analysis of kinematic data that can be measured by IVR systems could be helpful for monitoring the ongoing improvements of the patients.”

  1. Introduction

As a review article, it would be appropriate to include further information about the topic.

R: According to this comment we have expanded the Introduction with a paragraph about the USN (topic of this review) and better clarified the topic’s background: “USN can be defined as a deficit characterised by person’s failure to be aware of stimuli occurring on the side contralateral to the cerebral lesion, which result in  the inability to report and respond to stimuli  from this part of their visual field [4,5 and Robertson et al., 1993]. As USN is frequently associated with a lesion in the right hemisphere, people often have a left visual field deficit. This deficit is also often related to extrapersonal space [Matano et al. 2015], but it could also affect  personal space [Iosa et al. 2016]. . Further-more, USN is associated with poorer functional outcomes such as limited independence in daily tasks, increased risk of falls, longer hospital stays, and reduced likelihood of home discharge [6]. The patient with USN conventionally receives a pencil-paper training based on visual-scanning, reading and copying, copying of line drawings, and verbal description of a scene [Matano et al., 2015]. With ongoing advancements of technology, new high-tech innovations, such as Virtual Reality (VR), have been introduced to stroke rehabilitation and may offer a supplementary platform for promoting physical and cog-nitive recovery after stroke [7]. ”

About Virtual Reality we have also added the following paragraph

“VR should be more than a simple display of digital images as a computer videogame, but it should be able to bring the observer inside a 3D Virtual Environment that could be explored and that should respond in real time to the movements of the subject in a naturalistic way [Tieri et al., 2018]. Despite it, in clinical settings, often serious exergames at the basis of video-game based therapy are improperly referred to as “non-immersive” virtual reality. For the sake of clarity, and for being consistent with clinical scientific literature, we have used this terminology in this review [11]. ”

  1. Materials and Methods

Line 29: Delete “multiple” or change “several electronic databases” instead of “multiple electronic databases”.

R: Thanks for the suggestion. We change “multiple electronic databases” with “several electronic databases” in the current version of the manuscript.

3.Results

Poor quality of Figure 1, please add another of better quality.

R: Thanks for the suggestion. We attached a high-definition version of each Figure separately with respect to the manuscript.

Line 121: Review the total number of subjects included in the article, it comes out 211 instead of 199.

R: Thanks, for the observation, we revised the sample number in the revised manuscript.

Lines 136-138: “A synoptic table with complete studies' data is available in a comprehensive Table of supplementary materials (S2)”. I recommend to eliminate this Table from the supplementary material section and incorporate it into the text for a better reading and understanding.

R: Thanks for the suggestion, we have transferred Table 1 from supplementary materials to the main document.

Lines 140-142: “In the selected studies, the IVR training was characterized by a great variety of tasks (complete descriptions in Table 1). A description of each IVR task was reported in Table 2”. It does not correspond to the table incorporated in the text.

First of all, the Table has no legend and no table number. I recommend to insert a legend and the corresponding number. Second, there is only one table. It looks like two tables have been joined, but the explanation in the text has not been corrected.

R: Thanks for the observation. We have reported a detailed task description in Table 2, and made also the other changes required, also correcting some typos.

Line 149: Hagiwara and colleagues [25], Write Hagiwara et al. [25]

Thanks, done

Line 217: finally, for the other studies [22-24, 26], Incorporate study 17: [17, 22-24, 26]

 Thanks, done

4.Discussion

Line 226: 77 subjects with USN and 122 healthy subjects.

Review the number of healthy subjects

Thanks, done

Line 236: Tsirlin and colleagues in their review [35], Substitute for Tsirlin et al. [35] in their review

Thanks, done

Line 252: (Hagiwara et al., 2018). Replace with [25]

Thanks, done

  1. References

Line 359: review Hagiwara et al.

Thanks, done

As a review article, it would be appropiate to increase the number of references in certain sections (Introduction and Discussion).

Thanks for the suggestion, we have increased the references to provide an appropriate  background and support adequately the discussion in the revised manuscript.

Round 2

Reviewer 1 Report

There are 3 different aims described in this review: 

The present review aims to explore the use of immersive virtual reality (IVR) in the treatment of visual perception in unilateral spatial neglect (USN) after a stroke. Line 22 

The aim of the current review is to summarize most common features of the IVR systems used in neurorehabilitation and their effects on reducing the visual field and attention disorders related to unilateral spatial neglect after stroke. Line 90

This systematic comprehensive review aimed to investigate the effect of immersive virtual reality in the treatment of unilateral spatial neglect in stroke patients. Line 237

1.     The first aim would result in a usability evaluation and the outcome measure would only be visual perception in unilateral spatial neglect

2.     The second aim would be a summary of the most common features and then the effect of these common features on reducing the visual field and attention disorders.

3.     The third aim to investigate the effect of IVR in the treatment of USN.

Figure 1 PRISMA: 

Need info for every database that is – how many excluded from each database by title and then by abstract and then by full text. 

Do not just say how many were excluded from them all – must breakdown by database and by title etc (See above). This is due to the updated PRISMA guidance.

What do you mean by reports sought for retrieval?

Summary Table 1. could include Location of the study and drop out rates.

Discussion: Some reflection on the implementation of this intervention. How practical is it for stroke survivors to use IVR? Where would they use it? i.e. at home or in an outpatients department? How much supervision is required?

Author Response

Authors thanks to the reviewer for the valuable comments  and the opportunity to improve the manuscript.

Below the detailed responses: 

There are 3 different aims described in this review:

The present review aims to explore the use of immersive virtual reality (IVR) in the treatment of visual perception in unilateral spatial neglect (USN) after a stroke. Line 22

 The aim of the current review is to summarize most common features of the IVR systems used in neurorehabilitation and their effects on reducing the visual field and attention disorders related to unilateral spatial neglect after stroke. Line 90

 This systematic comprehensive review aimed to investigate the effect of immersive virtual reality in the treatment of unilateral spatial neglect in stroke patients. Line 237

  1. The first aim would result in a usability evaluation and the outcome measure would only be visual perception in unilateral spatial neglect

  1. The second aim would be a summary of the most common features and then the effect of these common features on reducing the visual field and attention disorders.

  1. The third aim to investigate the effect of IVR in the treatment of USN.

R: Thanks for the comment. The aim of this review is to analyze the efficacy of using IVR systems for the neurorehabilitation of patients with unilateral spatial neglect in reducing the deficits related to the attention in the visual field. This efficacy is strictly influenced by the usability of the IVR systems, the features of the planned interventions, those of the research protocols (including the sample sizes, the presence or not of a control group, if the study is a RCT and so on), and the selected outcome measures.

Figure 1 PRISMA:

Need info for every database that is – how many excluded from each database by title and then by abstract and then by full text.

Do not just say how many were excluded from them all – must breakdown by database and by title etc (See above). This is due to the updated PRISMA guidance.

What do you mean by reports sought for retrieval?

Thanks for the suggestion, we added the required information and edited step six in the flow diagram.

“Regarding the “Reports Sought for Retrieval” This is the number of articles obtained in preparation for full text screening. The number is calculated subtracting the number of excluded records from the total number screened.”

(see https://guides.lib.unc.edu/prisma)

 Summary Table 1. could include Location of the study and drop out rates.

R: Thanks for the suggestion, we added the location for each study in Table 1. About drop-out rates, in any studies were reported drop-outs. We added this information in the text:

 “The selected studies did not report any drop-out, all individuals finished the training, and post-intervention evaluations were analysed on the totality of the participants. ”

Discussion: Some reflection on the implementation of this intervention. How practical is it for stroke survivors to use IVR? Where would they use it? i.e. at home or in an outpatients department? How much supervision is required?

“In the analysed studies, no relevant severe adverse events have been reported. However, in other IVR experiences the literature reported slight symptoms such as dizziness, nausea, sore eyes and disorientation[38]. Tsirlin et al. [39] in their review highlighted some characteristics of VR technologies that should be considered for future VR applications in this field. The most important is the ergonomic aspect of VR tools, as people post stroke have specific needs that need to be considered, such as limited mobility [39]. Five of the studies included in this review [20, 24, 25, 31, 32] specified that the training was performed seated in a wheelchair or in a chair. Symptomatology, limitation to maintaining the upright position, and limitation in mobility suggests the use of IVR in a supervised environment such as a clinical setting.Despite the potential of in-home implementation, current evidence report an use in a controlled setting, safety assessment are needed before tested IVR technology in a domestic environment.”

Best regards

the authors